# Permeability of the Cyanotoxin Microcystin-RR across a Caco-2 Cells Monolayer

**DOI:** 10.3390/toxins13030178

**Published:** 2021-02-27

**Authors:** Jérôme Henri, Rachelle Lanceleur, Jean-Michel Delmas, Valérie Fessard, Antoine Huguet

**Affiliations:** Fougères Laboratory, French Agency for Food, Environmental and Occupational Health & Safety (ANSES), 35306 Fougères CEDEX, France; jerome.henri@anses.fr (J.H.); rachelle.lanceleur@anses.fr (R.L.); jean-michel.delmas@anses.fr (J.-M.D.); valerie.fessard@anses.fr (V.F.)

**Keywords:** microcystin-RR, Caco-2 cells, intestinal permeability

## Abstract

Microcystins (MCs) are toxins produced by several cyanobacterial species found worldwide. While MCs have a common structure, the variation of two amino acids in their structure affects their toxicity. As toxicodynamics are very similar between the MC variants, their differential toxicity could rather be explained by toxicokinetic parameters. Microcystin-RR (MC-RR) is the second most abundant congener and induces toxicity through oral exposure. As intestinal permeability is a key parameter of oral toxicokinetics, the apparent permeability of MC-RR across a differentiated intestinal Caco-2 cell monolayer was investigated. We observed a rapid and large decrease of MC-RR levels in the donor compartment. However, irrespective of the loaded concentration and exposure time, the permeabilities were very low from apical to basolateral compartments (from 4 to 15 × 10^−8^ cm·s^−1^) and from basolateral to apical compartments (from 2 to 37 × 10^−8^ cm·s^−1^). Our results suggested that MC-RR would be poorly absorbed orally. As similar low permeability was reported for the most abundant congener microcystin-LR, and this variant presented a greater acute oral toxicity than MC-RR, we concluded that the intestinal permeability was probably not involved in the differential toxicity between them, in contrast to the hepatic uptake and metabolism.

## 1. Introduction

Over the last century, due to increasing eutrophic conditions of freshwaters, there has been increasing incidence of massive proliferation of photosynthetic prokaryotic cyanobacteria [1]. Microcystins (MCs) are toxins produced by several cyanobacterial species that can affect humans through several routes of exposure. While exceptional, parenteral exposure can lead to death, as reported when MC-contaminated water was used for renal dialysis in Brazil, human exposure occurs mainly via the oral route through various sources, such as water, accumulating aquatic organisms, and dietary supplements [2,3,4,5]. Although the acute symptoms in humans due to oral exposure to MCs are not clearly identified [6,7], the toxicological profile of MCs raised some concern for human health [8].

MCs are a group of around 200 structurally different congeners of cyclic heptapeptides, with sizes ranging from 900 to 1100 Da [9]. Besides the global common structure, MC variants differ mainly by two _L_-amino-acids in positions 2 and 4, which were shown to affect their toxicity [10,11,12]. MCs induce toxicity through inhibition of serine/threonine protein phosphatases, in particular protein phosphatases 1 (PP1) and 2A (PP2A) [13,14,15], leading to the hyperphosphorylation of numerous proteins and, subsequently, to the disturbance of signal transduction and crucial cellular processes [16,17,18]. However, as the inhibition potency for PP1 and PP2A is very close between various MC variants, the toxicodynamics do not explain the differences in toxicity between them [8,19,20]. Rather, the toxicokinetic parameters seem to be affected by the variation of amino acids in positions 2 and 4 and therefore contribute to the toxicity of MC variants [21,22,23]. 

Among the MC variants, microcystin-RR (MC-RR) is the second most abundant congener [4]. It possesses two arginine amino acids in positions 2 and 4, making MC-RR more hydrophilic than other variants [4]. This cyanotoxin was shown to be absorbed through the gastrointestinal tract and distributed to different tissues [24], to induce toxicity in vivo [8,25,26], as well as in vitro in human intestinal epithelial Caco-2 cells [27,28], and also has an uptake in this cellular model [29]. As explained above, toxicokinetic parameters seems to contribute to the toxicity of MCs, and among these, intestinal permeability is a key parameter of oral toxicokinetics. To our knowledge, the intestinal permeability of MC-RR has not been evaluated. Therefore, we investigated the permeability of MC-RR across a Caco-2 cell monolayer. This cellular monolayer is a relevant in vitro model representing the human intestinal barrier and exhibits functional and morphological characteristics similar to enterocytes [30].

## 2. Results

Before exposure to MC-RR, all the monolayers had a trans-epithelial electrical resistance (TEER) above 250 ohm.cm² (ranging from 259 to 672 ohm.cm²) and therefore were considered suitable for the transport experiment. The integrity of Caco-2 cell monolayers was not affected by MC-RR, irrespective of the time and concentration, with TEER values ranging from 316 to 908 ohm.cm².

In apical (A) to basolateral (B) transport experiments, a decrease of the MC-RR amounts in the apical compartment (15–36% of the loaded amount) was observed within 1 h, irrespective of the initial concentration (Figure 1A,C,E,G). This decrease was followed by a plateau or a slight increase up to 24 h. Despite this large decrease of MC-RR amounts in apical compartments, only a small fraction of the toxin reached the basolateral compartment. For 1 and 10 µM MC-RR, the amount detected in the basolateral compartment was below the lower limit of quantification (LOQ) (<10 nM), irrespective of the exposure time, with the exception of 24 h for 10 µM MC-RR. We showed similar observations for 25 and 55 µM MC-RR during the first 2 h, with MC-RR amounts below the lower LOQ. Thereafter, the amount detected in the basolateral compartment remained rather low after 4 h exposure (on average 0.8% and 0.2% of the loaded amount) but reached 2.5% and 3.5% of the loaded amount after 24 h for 25 and 55 µM MC-RR, respectively. Therefore, apparent permeabilities (P_app_) were calculated for 25 and 55 µM MC-RR from 4 to 24 h exposure, and for 10 µM MC-RR, only after 24 h exposure. The values ranged from 4 to 15 × 10^−8^ cm·s^−1^, irrespective of the time and the loaded concentration, although no significant effect was observed (Table 1).

For B–A transport experiments, a decrease of MC-RR amounts in the basolateral compartment was observed within 1 h (on average 25% of the loaded amount), irrespective of the initial loaded concentration, followed by a plateau up to 6 h (Figure 1B,D,F,H). As for A–B transport experiments, only a small fraction of MC-RR reached the receiver compartment. During the first 4 h, for 1 and 10 µM MC-RR, the amount detected in the apical compartment was below the lower LOQ. After 6 h exposure, these amounts reached on average 0.9% and 0.3% of the loaded amount for 1 and 10 µM MC-RR, respectively. During the first hour, the amount detected in the apical compartment was also below the lower LOQ for 25 and 55 µM MC-RR. After 2 h exposure, these amounts remained rather low and were on average 0.05% and 0.03% of the loaded amount, and after 6 h, they reached 0.1% and 0.2% for 25 and 55 µM MC-RR, respectively. The corresponding calculated P_app_ ranged from 2 to 37 × 10^−8^ cm·s^−1^ (Table 1). There was no effect of the exposure time, and only an effect of the loaded concentration was reported after 6 h, with the P_app_ for 1 µM MC-RR being higher, as compared to that for 25 µM.

## 3. Discussion

As toxicity of MCs seems to be partly explained by toxicokinetic parameters, the aimof the present study was to investigatethe permeability of MC-RR across an in vitro intestinal barrier model, the Caco-2 cell monolayer. In our study, the A–B transport exhibited a rapid decrease in the amount of MC-RR in the apical compartment. This resulted in a significant uptake of MC-RR into Caco-2 cells, as shown using a fluorescent antibody [29]. However, crossing of MC-RR through the Caco-2 cell monolayer was low after 24 h exposure, with the relative amounts in the basolateral ranging from 2.5% to 3.5% of the initial amount. Irrespective of the experimental condition, these values corresponded to P_app_, ranging from 4 to 15 × 10^−8^ cm·s^−1^. The correlation proposed between the oral absorbed fraction in humans and the Caco-2 P_app_ distinguishes compounds of high permeability (P_app_ > 1 × 10^−6^ cm·s^−1^) from compounds with low permeability (P_app_ << 1 × 10^−6^ cm·s^−1^) [31,32]. Considering this assumption, our results suggest that MC-RR would be rather poorly absorbed orally. Similar results were observed for the B–A transport, characterized by a rapid decrease of MC-RR amounts in the donor compartment and low amounts crossing through the Caco-2 cell monolayer, leading to P_app_ values ranging from 2 to 37 × 10^−8^ cm·s^−1^. These values were similar, irrespective of the direction of the transport assay, and it appears that there was no involvement of an active efflux mechanism.

In comparison, permeability for microcystin-LR (MC-LR), the most abundant and studied MC congener, was also shown to be very low, with P_app_ values ranging from 2 to 5 × 10^−8^ cm·s^−1^ for A–B transport across the Caco-2 cell monolayer [33]. However, although MC-RR and MC-LR differ only in the _L_-amino acid in position 2 (arginine for MC-RR, leucine for MC-LR), MC-RR was ten-fold less toxic after intraperitoneal administration in mice [20,25] and also less toxic after oral administration to mice when compared to MC-LR [26]. This difference in toxicity was also reported in vitro using differentiated intestinal Caco-2 cells [27,28,34,35], as well as in primary human and rat hepatocytes [11,20]. As the inhibition potency for PP1 and PP2A established in vitro between these two variants is very close [19,20], and as they showed a similar intestinal permeability in vitro, the discrepancy in toxicity between MC-LR and MC-RR cannot likely be explained by toxicodynamics, but rather by characteristics other than intestinal permeability, such as absorption, distribution, metabolism, and excretion.

Due to their structure, MCs are unable to cross cell membranes via passive diffusion, but rather require transporters. Among them, it is well established that organic anion transporting polypeptides (OATPs) are involved in the cellular uptake of MCs [36,37,38]. More precisely, OATP1B1 and OATP1B3 have been demonstrated as the two main isoforms involved in the transport of MCs into cells [38]. Several studies reported that MC-RR was less toxic than MC-LR in different cell lines over-expressing OATP1B1 and/or OATP1B3, suggesting a lower cellular uptake of MC-RR through these two OATPs when compared to MC-LR [20,37,39]. This was explained by a lower affinity of MC-RR for these two OATPs when compared to MC-LR [40]. These two OATPs are mainly expressed in the liver, while their presence in the intestine was not detected [41]. In consequence, there was a lower toxicity of MC-RR in primary human and rat hepatocytes when compared to MC-LR [11,20]. Moreover, the absence of these two transporters in the intestine is in accordance with the very low and similar intestinal permeabilities of MC-LR and MC-RR determined in our study and that of Henri et al [33]. Nevertheless, the decrease of MC-LR and MC-RR amounts in the apical compartment suggested an involvement of other OATP isoforms in this uptake into enterocytes, probably OATP3A1 and OATP4A1, as suggested by Zeller et al. [29], as these two OATPs are expressed in the intestine [41].

The metabolism could also explain the discrepancy in toxicity between MC-LR and MC-RR. The detoxification of MC-LR and MC-RR occurs mainly through glutathione conjugation, catalyzed by glutathione-S-transferases, leading to conjugates with minimal residual inhibitory activity, with respect to the parent compound [23,42,43,44,45]. However, MC-RR is more efficiently conjugated than MC-LR, and this greater biotransformation of MC-RR could explain its lower toxicity compared to MC-LR [46]. Moreover, as the two congeners have similar uptake profiles into enterocytes, and this differential detoxification could also explain the lower toxicity of MC-RR when compared to MC-LR observed in differentiated intestinal Caco-2 cells [27,28,34].

In summary, MC-RR presented a very low intestinal permeability. Its low P_app_ values were similar to that of MC-LR. Therefore, the lower acute oral toxicity of MC-RR when compared to that of MC-LR is probably driven by a differential cellular uptake occurring in hepatocytes, but also a differential cellular metabolism occurring in hepatocytes and enterocytes. Consequently, mechanistic studies to determine and compare such parameters for several MCs variants are necessary to establish toxic equivalent factors and to deal with risk assessment following oral exposure to MCs.

## 4. Materials and Methods 

### 4.1. Chemicals

Cell culture products (culture medium, non-essential amino acids, penicillin, streptomycin, and fetal calf serum) and Hank’s balanced salt solution (HBSS) were purchased from Gibco (Invitrogen, Cergy-Pontoise, France). MC-RR (purity certified higher than 95% by an HPLC method) was supplied by Novakits (Nantes, France). Methanol was purchased from Fisher scientific (Val de Reuil, France).

### 4.2. Cell Culture

Caco-2 cells were obtained from the American Type Culture Collection (HTB-37, LGC Standards, Molsheim, France) and used at passages 32–34. The cells were maintained in culture medium (minimum essential medium containing 5.5 mM D-glucose, Earle’s salts, and 2 mM L-alanyl-glutamine (MEM GlutaMax)), supplemented with 10% fetal calf serum, 1% non-essential amino acids (glycine, L-alanine, L-asparagine, L-aspartic acid, L-glutamic acid, L-proline, and L-serine), 50 IU/mL penicillin, and 50 µg/mL streptomycin at 37 °C in an atmosphere containing 5% CO_2_. For transport experiments, Caco-2 cells were seeded at 6 × 10^4^ cells/cm^2^ on polyester membrane inserts (0.4 µm pore size, 12 mm diameter), purchased from Corning (Corning, NY, USA). Culture medium was changed three times a week, and cells were used on days 25–27 post-seeding.

### 4.3. Bidirectionnal Transport Experiments

Experiments were performed as previously described [47], with a few modifications, described as follows. Four solutions of MC-RR (1, 10, 25, and 55 µM) were prepared in the transport buffer (HBSS, 4-(2-Hydroxyethyl)piperazine-1-ethanesulfonic acid (HEPES) 5 mM, pH 7.4) in glass vials and loaded into the donor compartments: 300 µL in compartments A (for A–B transport experiments) and 1000 µL in compartment B (for B–A transport experiments). The final concentration of methanol in the transport buffer for all MC-RR solutions was set at 1.3%, which was previously shown not to alter the integrity of Caco-2 cell monolayers. In a preliminary experiment, the binding of MC-RR to plastic was checked and considered negligible (data not shown). At different time points (1, 2, 4, 6, and 24 h for A–B transport experiments and 1, 2, 4, and 6 h for B–A transport experiments), the whole volume of each compartment was harvested and stored in glass vials at −20 °C until analysis. Due to the toxicity of methanol 1.3% when applied in compartment B for 24 h, this time point was not performed for B–A experiments. Only Caco-2 cell monolayers with a TEER > 250 ohm.cm² before and after transport experiments were used. With the historical data generated in our laboratory, a TEER value of 250 ohm.cm² corresponds to a P_app_ of lucifer yellow of 6 × 10^−7^ cm·s^−1^. These threshold values correspond to those preconized by the European Center for the Validation of Alternatives Methods [48,49]. Four independent experiments were performed.

### 4.4. Analytical Method

An analytic method was developed and validated for this study. Briefly, chromatographic separations were performed on an Accela liquid chromatography U-HPLC system (ThermoFisher, Bremen, Germany), equipped with a Zorbax RX C8 column (150 × 2.1 mm; 5 µm particle size) and an Eclipse guard column (10 × 2.1mm, 5 µm particle size), from Interchim (Montluçon, France). The column oven temperature was set to 30 °C, the flow rate used was 200 µL/min, and the injection volume was 20 µL. The mobile phase consisted of formic acid 0.1% in water and acetonitrile. The elution was isocratic with 60% of formic acid 0.1% in water and 40% of acetonitrile. The data acquisition time was 10 min. Mass spectral analysis was carried out on LTQ-Orbitrap mass spectrometer XL MS (Thermofisher) with an electrospray ionization probe and operated in the positive ion mode. The instrument was calibrated using the manufacturer’s calibration solution (consisting of caffeine, the tetrapeptide H-Met-Arg-Phe-Ala-OH (MRFA), and Ultramark) to reach mass accuracies in the 1–3 ppm range. Parameters of the ion source were as follows: capillary voltage 65 V, ion spray voltage 3.8 kV, tube lens 120 V, capillary temperature 300 °C, sheath gas flow 50 (arbitrary units), auxiliary gas flow 20 (arbitrary units), and sweep gas 0 (arbitrary units). Nitrogen was used as the sheath and auxiliary gas in the ion source. The analysis was performed according to a full scan Fourrier transform mass spectrometry from m/z 500–1100 at a resolving power of 60,000 (full width at half maximum).

In order to determine the accuracy and the lower LOQ of this specifically developed method, a validation was performed according to the total error approach [50] and performances of the method were synthetized using eNoval (version 1.1a, Arlenda, Liège, Belgium) in an accuracy profile (Figure 2). The variation in accuracy for the quantification of MC-RR in transport buffer was always below 20% (acceptance limits), as shown in Figure 2. The lower LOQ of the method was 10 nM, and the upper LOQ was 1000 nM. The samples with concentrations above the upper LOQ were diluted before analysis, while the samples with concentrations below the lower LOQ were removed from the analysis.

### 4.5. Data Analysis

The P_app_ was calculated using the following equation:Papp = (dQ / dt) / (S × C0)(1)
where dQ/dt, S, and C_0_ represent, respectively, the amount of MC-RR transported within a given time period, the surface area of the Caco-2 cells monolayer, and the initial concentration of MC-RR loaded into the donor compartment. Statistical analyses were performed using GraphPad Prism software (version 5.0; GraphPad Software Inc., La Jolla, CA, USA), and a one-way analysis of variance was performed. When the time or concentration effect was significant (*p* < 0.05), the values were compared using Bonferroni’s test. Differences were declared significant at *p* < 0.05. The values presented are mean ± SEM.

## Figures and Tables

**Figure 1 toxins-13-00178-f001:**
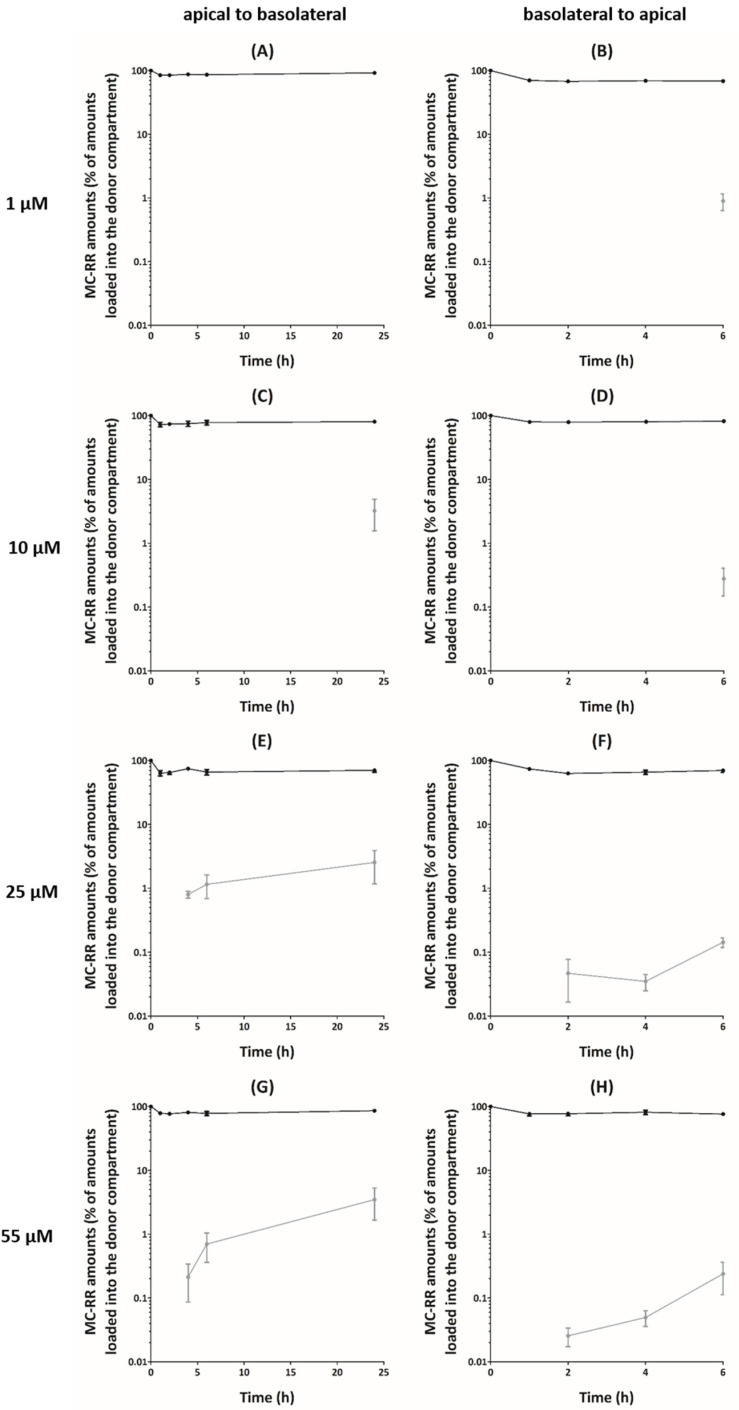
Relative amounts of microcystin-RR (MC-RR) following exposure of Caco-2 cell monolayers. Transport experiments were performed from apical to basolateral compartments (**A**,**C**,**E**,**G**) and from basolateral to apical compartments (**B**,**D**,**F**,**H**). Values of MC-RR amounts in the compartments are presented as mean ± SEM and expressed as percentage of amounts loaded into the donor compartment. Donor compartment: black dots and lines. Receiver compartment: grey dots and lines. Missing data points mean that MC-RR was not detected. Four independent experiments were performed.

**Figure 2 toxins-13-00178-f002:**
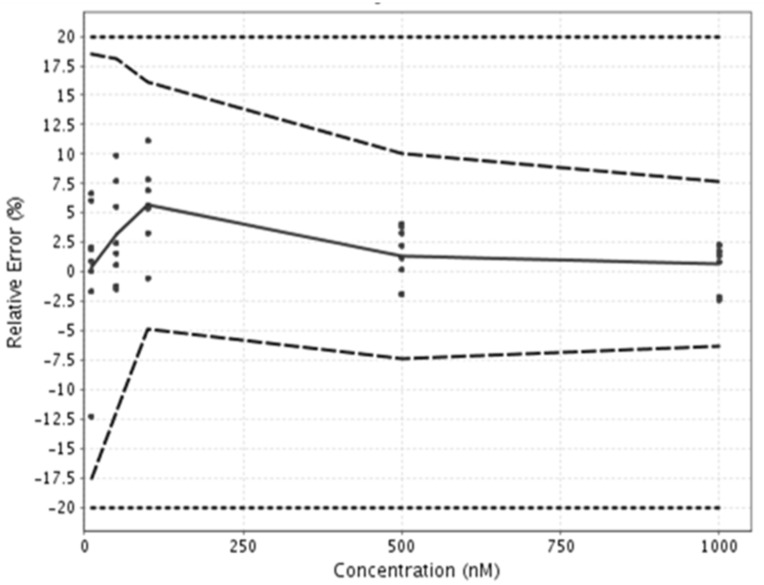
Accuracy profile of the analytical method used for microcystin-RR quantification. The points show validation data, the plain line denotes trueness of the method, the dashed lines denote precision of the method, and the dotted lines denote the acceptance limits.

**Table 1 toxins-13-00178-t001:** Apparent permeabilities (P_app_) of microcystin-RR, following exposure of Caco-2 cell monolayers. Transport experiments were performed from apical to basolateral compartments (A–B) and from basolateral to apical compartments (B–A). P_app_ values are presented as mean ± SEM and expressed in 10^−8^ cm·s^−1^. Four independent experiments were performed. Not determined (ND); not applicable (NA).

Transport	Loaded Concentration (µM)	Time (h)				
		1	2	4	6	24
A–B	1	NA	NA	NA	NA	NA
	10	NA	NA	NA	NA	10.0 ± 5.2
	25	NA	NA	14.8 ± 1.8	14.3 ± 5.7	7.9 ± 4.2
	55	NA	NA	4.0 ± 2.4	8.7 ± 4.2	10.8 ± 5.6
B–A	1	NA	NA	NA	37.0 ± 10.7	ND
	10	NA	NA	NA	11.5 ± 5.3	ND
	25	NA	5.8 ± 3.8	2.2 ± 0.6	5.9 ± 1.0	ND
	55	NA	3.2 ± 1.0	3.1 ± 0.8	9.9 ± 5.2	ND

## Data Availability

The data presented in this study are available on request from the corresponding author.

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
