# Peer review of "Permeability of the Cyanotoxin Microcystin-RR across a Caco-2 Cells Monolayer"

_toxins, 2021, doi:10.3390/toxins13030178_

Round 1

Reviewer 1 Report

In the manuscript the authors investigated a permeability of cyanotoxin MC-RR across a differentiated intestinal Caco-2 cells monolayer.

There are several comments.

It should be emphasized in the text of the manuscript:

1) What scientific hypothesis is being tested?

2) Why was this particular model chosen?

3) What is the scientific novelty of the study?

Unfortunately, the scientific significance of the article is not obvious.

4) Can the results of in vitro experiments be extrapolated to an in vivo system? How valid are the conclusions drawn?

Minor comments:

(1) Results

There are only One Table and one Figure in this section.

Figure 1. Donor compartment: black dots and lines. Why deviations were not shown for these results?

Table 1. Why four NDs (not determine) are in this Table (Transport B-A)? Why the authors did not determine these values? Please, explain.

(2) Material and Methods

Line 168. “Cell culture products”. Please, indicate which ones?

Lines 177. “1% non-essential amino acids”. Please, indicate which ones?

Why 50 IU/mL penicillin and 50 μg/mL streptomycin were added? Please, explain this point to readers.

Reviewer 2 Report

Dear authors, 

the paper dedicate to a hot topic of recent science is well-designed and relevantly well presented in an understandable language. I support its publication in Toxins.

Reviewer 3 Report

The work described in "Permeability of the Cyanotoxin Microcystin-RR across a Caco-2 2 Cells Monolayer" is well designed and constructed. All the assays are well conducted and well justified, but the soundness of the whole worka is a little too small to constitute a full article in an international journal. Perhaps the authors could include some additional experiments on the differences between these two congeners at the hepatic level as hypothesised in the conclusions, to confirm this theory and give more value to the article.

Reviewer 4 Report

The algal bloom in meso- and eutrophic water bodies and the deterioration of water quality caused by this phenomenon are becoming serious problems in many countries of the world. The most dangerous and common bloom is the mass growth of cyanobacteria and, as a result, the appearance of toxins in the water the concentration of which exceeds the values dangerous for people and animals.

One of the key problems in the modern toxicology and biomedicine is to understand the process of delivery of toxins to an organ or target cells. In this regard, it is necessary to objectively assess the behaviour of toxins in the gastrointestinal tract. Since the 1990s, when the possibility of assessing the permeability using in vitro cell models was established for the first time, this direction has been actively developing. Among cell models, MDCK, HT29-MTX, and TC-7), the colon adenocarcinoma cell culture (Caco-2) has become the most widely used for this purpose [Le Ferrec et al., 2001]. Based on the foregoing facts, the problems raised in the article are relevant and interesting for fundamental science and have applied significance.

At the same time, there are some remarks:   

  1. I think that the Authors should have also analysed other variants of microcystins, for example, MC-YR, which would allow them to compare the most common and structurally close congeners and possibly shed light on the influence of other factors in addition to the permeability of the monolayer.
  2. In my opinion, the authors should correct English, desirably by a native speaker.
  3. I would recommend the Authors to use the test with low permeability markers (for instance, lucifer yellow, rhodamine, mannitol, or PEG-400) simultaneously with the measurement of transepithelial electrical resistance.

Round 2

Reviewer 3 Report

The work is well described and ejecuted. It is small but well conducted